# Low-Volume (0.3 mL/kg) Ropivacaine 0.5% for a Quadratus Lumborum Block in Cats Undergoing Ovariectomy: A Randomized Study

**DOI:** 10.3390/vetsci12060524

**Published:** 2025-05-28

**Authors:** Chiara Di Franco, Camilla Cozzani, Iacopo Vannozzi, Angela Briganti

**Affiliations:** 1Department of Veterinary Sciences, Veterinary Teaching Hospital “Mario Modenato”, via Livornese snc, San Piero a Grado, 56121 Pisa, Italy; c.cozzani3@studenti.unipi.it (C.C.); iacopo.vannozzi@unipi.it (I.V.); angela.briganti@unipi.it (A.B.); 2Institute of Clinical Physiology, CNR San Cataldo Research Area, 56121 Pisa, Italy

**Keywords:** loco-regional anesthesia, feline, ultrasound-guided, pain management

## Abstract

In recent years, the use of locoregional anesthesia techniques has gained popularity in veterinary surgeries. Our study aimed to assess the effectiveness of a quadratus lumborum block (QLB) in cats getting spayed. A total of 22 cats were premedicated with a mix of sedatives and opioids and randomly assigned to receive either the QLB with ropivacaine 0.5% (the Ropi group) or a saline solution (the NaCl group), with the block administered via a lateral approach. In cases of pain, a fentanyl IV bolus was administered, or if the bolus was insufficient, a fentanyl IV infusion was started. Clinical parameters such as heart rate (HR), blood pressure, capillary refill time, ECG rhythm, EtCO_2_, temperature, oxygen saturation, fraction of expired isoflurane, and spirometry were monitored at specific surgical intervals, including the need for and dosage of analgesics and vasoactive medications. Notably, seven out of eight cats in the NaCl group required a fentanyl bolus, while only two out of ten in the Ropi group did. This research indicates that the ultrasound-guided bilateral quadratus lumborum block with ropivacaine effectively provides adequate pain relief during ovariectomy in cats.

## 1. Introduction

Elective surgical ovariectomy is one of the most common surgical procedures in veterinary medicine. This is due to several factors, including the prevention of reproductive tract diseases, like mastitis, metritis, and pyometra, as well as mammary tumors. Additionally, it helps mitigate undesirable behaviors linked to hormonal changes, aids in population control, and reduces the number of stray animals [1]. In human medicine, the concept of Enhanced Recovery After Surgery (ERAS) has emerged and is increasingly being adopted in veterinary practices [2]. Initially introduced by Prof. Kehlet in the late 1990s for colorectal surgery, ERAS has been successfully implemented across various surgical procedures, demonstrating a reduction in morbidity and mortality rates. This approach encompasses a series of evidence-based interventions applied throughout the preoperative, intraoperative, and postoperative phases, utilizing a multidisciplinary strategy. A key aspect of ERAS is the emphasis on effective perioperative analgesia, particularly through locoregional anesthesia, which helps minimize opioid consumption and enhances recovery after surgery [3]. In recent years, interfascial blocks have become increasingly important for managing acute postoperative pain [4]. The transversus abdominis plane (TAP) block and the QLB are two of the most used regional anesthesia techniques in both human and veterinary medicine [2,5,6,7]. These two blocks operate through distinct mechanisms: the TAP block is specifically designed to target and desensitize the thoracolumbar ventral rami of the spinal nerves, providing somatic analgesia to the abdominal wall and peritoneum; the QLB involves the administration of local anesthetic within the fascia that envelops the quadratus lumborum muscle. This strategic placement allows the anesthetic to affect the thoracolumbar ventral rami of the spinal nerves and the sympathetic trunk, which are crucial for the somatic and visceral innervation of the abdominal region [3]. The mechanism of action remains inadequately understood; however, it seems that local anesthetics, diffusing through the thoracolumbar fascia and endothoracic fascia, reach the paravertebral space [2]. One study indicated that contrast material administered at the quadratus lumborum plane did not extend into the paravertebral space, while fluid injected into the paravertebral area failed to reach the quadratus lumborum region [8]. This observation supports the hypothesis that visceral analgesia may arise from the diffusion of anesthetic to the celiac ganglion or along the sympathetic trunk via the splanchnic nerves [2]. Another possible explanation could involve the dense network of sympathetic nervous system neurons located on the surface of the thoracolumbar fascia, which contains both mechanoreceptors and nociceptive sensory receptors integral to the processes of acute and chronic pain. Therefore, the analgesia could also result from the blockade of these receptors [2].

In veterinary medicine, several cadaveric studies have described the application of the QLB technique in dogs [9,10,11,12] and cats [13,14,15], as well as the spread of anesthetic in the tissues. However, there are few clinical studies describing the use of this technique for abdominal procedures in dogs [7,16,17,18,19] and cats [20,21,22,23]. All the studies conducted in cats evaluated the efficacy of the QLB in managing perioperative pain due to ovariectomy. Ovariectomy is a valuable model for evaluating the effectiveness of the QLB, as the surgery naturally generates visceral pain due to tissue manipulation. Furthermore, the anatomical proximity of the quadratus lumborum muscle to the surgical site provides critical information regarding the block’s efficacy in a controlled environment. The majority of studies performed in cats used a high volume (0.4–0.5 mL/kg) of bupivacaine 0.25% [22,23] or ropivacaine 0.4% [21] and the only one in which a low volume of 0.3 mL/kg was used showed a higher request of intraoperative analgesia compared to the high-volume groups (0.5 mL/kg of bupivacaine 0.2%) [20].

Our hypothesis is that QLB with a low volume (0.3 mL/kg) but a higher concentration of ropivacaine (0.5%) could be effective in reducing the intraoperative analgesia requirement in cats undergoing ovariectomy. Thus, the primary aim of this study was to evaluate the efficacy of the quadratus lumborum block in cats undergoing laparotomic ovariectomy with ropivacaine 0.5% with a low volume of 0.3 mL/kg in comparison to a control group.

## 2. Materials and Methods

Twenty-two female European mix breed cats of different ages, and weights, undergoing ovariectomy from February 2024 to June 2024 at the Veterinary Teaching Hospital “Mario Modenato” of the University of Pisa, were enrolled in this randomized prospective blind study.

This study was previously approved by competent authorities (OPBA) of the University of Pisa n.9/2024. The PetSORT guidelines were applied [24].

The owners of the animals signed informed consent for both the anesthesia and the surgical procedure. Inclusion criteria consisted of clinically healthy cats scheduled for ovariectomy. Exclusion criteria included the need for repeated premedication doses or the presence of irritation and/or infection at the nerve block site during skin preparation (trichotomy and disinfection).

On the day of the surgery, each cat received an intramuscular premedication with 15 mcg/kg dexmedetomidine (Dexdomitor^®^, Orion Pharma, Milan, Italy), 0.2 mg/kg methadone (Semfortan^®^, Dechra, Milan, Italy) and 2 mg/kg alfaxalone (Alfaxan^®^ Multidose, Zoetis, Rome, Italy).

The cats were then placed under oxygenation with a mask of the appropriate size. Essential monitoring, such as electrocardiogram and non-invasive blood pressure measurement, were also performed. A venous catheter (Delta Ven^®^, Deltamed Medical Devices, Mantua, Italy) was then positioned in one of the cephalic veins, from which lactated Ringer was administered at a dose of 2 mL/kg/h.

Alfaxalone (Alfaxan^®^ Multidose, Zoetis, Rome, Italy) was used to induce the animals intravenously, to effect to achieve orotracheal intubation. The endotracheal tube was connected to a rebreathing system and anesthesia was maintained with Isoflurane (Isoflurane–Vet, Merial, Italy) in an oxygen–air mixture with a fraction of inspired oxygen (FiO_2_) of 0.7. An arterial catheter (Delta Ven^®^, Deltamed Medical Devices, Mantua, Italy) was inserted at the metatarsal level during the preparation phase for invasive blood pressure monitoring. A second venous catheter was placed for potential emergency drug administration during surgery. All patients received prophylactic ampicillin (Vetamplius, Fatro Spa, Milan, Italy) at a dose of 22 mg/kg IV prior to induction. Subsequently, a lateral–ventral trichotomy of the abdomen was performed, both for the execution of the QLB and for the preparation of the surgical site. Before the premedication, cats were randomly (https://www.random.org, accessed on 10 May 2025) divided into two groups of eleven each: the Ropi group, where the block was performed with Ropivacaine (Ropivacaine Kabi 7.5 mg/mL, Fresenius Kabi, Italy) at 0.5% at 0.3 mL/kg per side, and the NaCl group, which received an equal volume (0.3 mL/kg) of saline solution. All blocks were performed by the same anesthetist (C.D.F.) using an along-the-visual-axis technique [25] with a dedicated veterinary ultrasound system (Sonosite S II Veterinary Ultrasound System, Fujifilm Italia S.p.a., Milan, Italy) and a linear ultrasound probe (HFL50, 15–6 MHz Linear Transducer). The QLB was performed as described by Garbin et al. using an 85 mm spinal needle for nerve blocks (Visioplex, Vygon Italia S.r.l., Padua, Italy). The anesthetists in charge of intraoperative monitoring (C.C.) were unaware of the protocol used (Ropi or NaCl group). No other study to date has used this specific combination of volume, concentration, and drug (ropivacaine 0.5% at 0.3 mL/kg) in a clinical feline model with blind observers.

The cats were then transported to the operating room and connected to an anesthetic workstation (Avance CS^2^, GE Healthcare, Bensalem, PA, USA). Volume-controlled ventilation with a Tidal Volume of 10 mL/kg was set to maintain end-expiratory CO_2_ (EtCO_2_) between 35 and 45 mmHg. During surgery, heart rate (HR), arterial blood pressure, capillary refill time (CRT), ECG rhythm, EtCO_2_, temperature (T), oxygen saturation (SpO_2_), fraction of expired isoflurane (Fe’Iso) and spirometry were monitored every five minutes (Avance CS^2^ Pro, GE, Milan, Italy) and recorded at specific surgical time points (Table 1).

The procedures were conducted by various student surgeons, who were not specialists but were consistently overseen by the supervising professor of surgery. If nociception was suspected based on an increase in heart rate and/or mean arterial pressure (MAP) greater than 15% compared to the previous value, an IV bolus of fentanyl at 1 mcg/kg (Fentadon, 50 mcg/mL, Dechra, Milan, Italy) was administered. If parameters did not return to baseline following the fentanyl bolus, a variable CRI of fentanyl was initiated at 2 mcg/kg/h. Any episodes of hypotension (MAP < 60 mmHg) were initially treated with a bolus of Ringer’s lactate (2 mL/kg IV). Patients that remained hypotensive and failed to respond to fluid bolus therapy, defined as an increase in arterial blood pressure or decrease in heart rate by at least 10% of pre-bolus values, were started on a variable-rate IV infusion of norepinephrine 0.05 mcg/kg/min with incremental doses of 0.05 mcg/kg/min until normotension was restored. In cases of hypotension (MAP < 60 mmHg) with concurrent bradycardia (HR < 100bpm), atropine at the dose of 0.02 mg/kg IV (atropine sulphate, A.T.I., Bologna, Italy) was administered. At the end of surgery, isoflurane administration was stopped, and when the cats returned to spontaneous ventilation, mechanical ventilation was stopped. Each cat was then moved to the recovery room and tracheal extubation was performed once the swallowing reflex returned. The duration of anesthesia (from induction to extubation) and the duration of surgery were recorded. Each patient received 2 mg/kg of robenacoxib (Onsior^®^, Novartis Farma S.p.A, Milano, Italy) subcutaneously. All cats were released within hours following surgery once their health was deemed satisfactory; consequently, a post-operative pain assessment could not be performed.

### Statistical Analysis

The sample size was calculated using Clincalc.com. Based on a previous study in which the QLB was effective in the 84% of cats undergoing ovariectomy and the 20% of cats in the control group did not required intraoperative rescue analgesia [20], with an alfa error of 0.05 and a power of 80%, the minimum number of cats to be enrolled resulted 8 for each group; we decided to increase the sample size to 11 each group to account for potential dropouts.

The data were analyzed for normal distribution using a D’Agostino and Pearson tests and are expressed by mean and standard deviation. The trend of the parameters in the different times monitored for each group was analyzed using ANOVA for repeated data with a Dunnett test as post hoc, while the evaluation between the two groups was performed using a Student’s *t* test. The comparison between the groups for the number of subjects who required rescue analgesia was evaluated using Fisher’s exact test. Values of *p* < 0.05 were considered significant.

## 3. Results

The study enrolled 22 cases, of which 4 were excluded because they required repetition of the premedication dose due to extreme agitation (Figure 1).

Of the 18 cases, 10 were included in the Ropi group and 8 in the NaCl group. No differences were found regarding the weight or age of the cats, nor regarding the duration of anesthesia and surgery between the two groups (Table 2). The quality of recovery was excellent in all patients.

Regarding hemodynamic parameters, the heart rate in the Ropi group was significantly higher at time T5 than at T0 (*p* = 0.03), while in the NaCl group, a significant increase was highlighted compared to T0 from time T4 to time T7 (*p* < 0.01) (Figure 1); no differences were detected between the two groups regarding HR, even though the values of the Ropi group were constantly lower. The other parameters, including arterial blood pressure, temperature, (CRT), ECG rhythm, EtCO_2_, SpO_2_, Fe’Iso, and spirometry did not show significant differences either within the groups or between the two groups, even though the values of the NaCl group were higher (Figure 2).

Regarding the need for intraoperative rescue analgesia, the NaCl group showed a significantly higher number of subjects (7/8) that required a bolus of fentanyl and subsequently continuous infusion of fentanyl compared to the Ropi group (2/10) (*p* = 0.01) (Figure 3).

In the Ropi group, one out of ten patients received rescue analgesia at T1 and one out of in ten at T2. In the NaCl group, two out of eight received the fentanyl bolus at T1 and five out of eight at T2 (Table 3).

Furthermore, in the Ropi group, 7 out of 10 animals required one bolus of atropine (20 mcg/kg IV) (7/10 at T1), while in the NaCl group, 4 out of 8 required it (1/8 at T1 and 3/8 at T2), and no significant difference was found between the two groups.

## 4. Discussion

The present study demonstrated that the QLB, using a low volume 0.3 mL/kg of ropivacaine at 0.5%, was effective in providing perioperative analgesia in cats undergoing ovariectomy. The study was conducted within a multimodal analgesia plan, where the premedication protocol included a combination of dexmedetomidine and methadone. From a hemodynamic point of view, a significant increase in heart rate was detected at time T5 (ligation and removal of the second ovary) compared to time T0 (pre-surgery) in the Ropi group. In the NaCl group, a significant increase in heart rate was observed from time T4 (search for the second ovary) to time T7 (skin suturing) compared to time T0 (pre-surgery). However, vasoactive drugs (atropine) were used in many of the patients, and it was administered frequently in both the Ropi group (7 out of 10 subjects) and the NaCl group (4 out of 8 subjects). Atropine is the drug of choice for managing bradycardia and hypotension, which can occur during deep sedation or general anesthesia and may lead to syncopal episodes [26]. The systemic administration of atropine generally results in an increase in both heart rate and blood pressure [27]. In our study, atropine was primarily administered before the beginning of surgery (T1) in the Ropi group, which may explain the increase in heart rate observed in patients who underwent the ropivacaine QLB. Furthermore, although there was no significant difference between the two groups, the lower use of atropine in the NaCl group (4 out of 8 cases versus 7 out of 10 in the Ropi group) could be attributed to an increase in heart rate and blood pressure due to the nociceptive stimulus, which made subsequent administration of the drug unnecessary. The study included an analgesia protocol for pharmacological intervention in the management of nociception, which involved administering a fentanyl bolus of 1 mcg/kg and, if needed, a continuous fentanyl infusion with a dosage ranging from 2 to 15 mcg/kg/h depending on the patient’s requirement. In the NaCl group, fentanyl was administered in 7 out of 8 cases, while in the Ropi group, the fentanyl bolus was given in 2 out of 10 cases.

These results are in line with the previous studies: the quadratus lumborum block provides both somatic and visceral analgesia at the abdominal level [16].

In veterinary medicine, several cadaveric studies have been conducted on dogs [9,10,11,12] and on cats [13,14,15], analyzing the spread of the anesthetic in various approaches to the technique. However, clinical studies are limited [7,16,17,18,19,20,21,22,23]. In a study involving 10 dogs undergoing ovariohysterectomy, the quadratus lumborum block was performed using 0.5% bupivacaine mixed with contrast medium at a 1:1 ratio per side. The spread of the injected solution was assessed via CT scan [7], and the study reported that only one dog required rescue analgesia during the manipulation of the ovarian ligament. The spread observed in the CT scan was consistent with cadaveric studies in which the same technique had been used [11]. In a more recent study [20], 48 cats undergoing ovariectomy were recruited for a bilateral ultrasound-guided quadratus lumborum block using 0.2% bupivacaine, with the lateral approach described by Garbin et al. [10]. The cats were divided into three groups: the first received a low-volume quadratus lumborum block (QLB-LV 0.3 mL/kg), the second received a high-volume block (QLB-HV 0.5 mL/kg), and the third, a control group, received a saline block. In the QLB-LV group, fentanyl was required in 10 out of 16 cases; in the QLB-HV group, it was needed in 2 out of 16; and in the saline group, it was administered in 13 out of 16 cases [20]. In our study, a low-volume (0.3 mL/kg) quadratus lumborum block was performed using the same approach [10], and fentanyl was administered in 2 out of 10 cases in the Ropi group. However, the two studies differ in the local anesthetics used and their concentrations (0.2% bupivacaine versus 0.5% ropivacaine) even if both studies followed a multimodal analgesic approach. Lazzarini and colleagues included in the premedication dexmedetomidine, methadone, and ketamine; in our protocol, however, ketamine was not included to avoid an additional analgesic effect. Despite this, fentanyl was used in only 20% of cases in the Ropi group. Our study is in line with the results reported by Paolini et al., who used higher volumes (0.4 mL/kg) and lower concentrations (0.4%) of ropivacaine. This suggests that, although the volume is reduced, the effectiveness of the intraoperative block remains consistent. Two more studies have been published in cats by do Santos and colleagues, but in both studies, a higher volume (0.4 mL/kg) and lower concentration (0.25%) of bupivacaine were used [22,23]. In the first study, the authors demonstrated that the QLB was efficacious and had less impact on motor function and hemodynamics than sacrococcygeal epidural anesthesia. In the second study, the authors demonstrated that the QLB group required less intraoperative analgesia than the control group, and this result is consistent with our findings. The two studies are comparable because the premedication and anesthetic protocol were similar, which suggests that 0.5% ropivacaine at a low volume has a similar intraoperative analgesic effect to 0.4 mL/kg of 0.25% bupivacaine. However, it is important to highlight that in do Santos’ study, only experienced surgeons performed the surgeries, whereas in our study, the surgeons were novices. Therefore, additional studies are required to compare the efficacy of ropivacaine and bupivacaine at various concentrations and volumes. To the best of our knowledge, this is the first study to show that a low-volume dose of 0.3 mL/kg high-concentration (0.5%) ropivacaine can achieve intraoperative analgesia comparable to traditional higher-volume protocols. These findings support a more volume-efficient approach in feline anesthesia, offering effective pain management while minimizing drug volume and potential risks.

This study has several limitations. First, it was conducted on a sample size, finalized only to evaluate the intraoperative analgesic request. As a result, some differences between the two groups may not have been detected. Additionally, there was considerable variability in the surgical procedures as they were conducted by less experienced surgeons. Lastly, the animals were discharged within hours of the operation after confirming their good health status, which restricted the opportunity for postoperative monitoring and hindered the evaluation of postoperative pain or variations in pain levels between the two groups.

## 5. Conclusions

In conclusion, the present study confirmed that ultrasound-guided bilateral quadratus lumborum block with a low volume of 0.3 mL/kg and 0.5% ropivacaine is effective in providing perioperative analgesia in cats undergoing ovariectomy. To date, no other study has evaluated this specific combination of drug, concentration, and volume in a clinical feline model using blind observers. Unlike previous findings with bupivacaine at the same volume, these results suggest that a higher anesthetic concentration can compensate for lower volume, supporting a more conservative, yet effective, analgesic approach in feline anesthesia. Additional clinical studies are necessary to evaluate the postoperative effects and make eventual comparisons with the use of bupivacaine.

## Figures and Tables

**Figure 1 vetsci-12-00524-f001:**
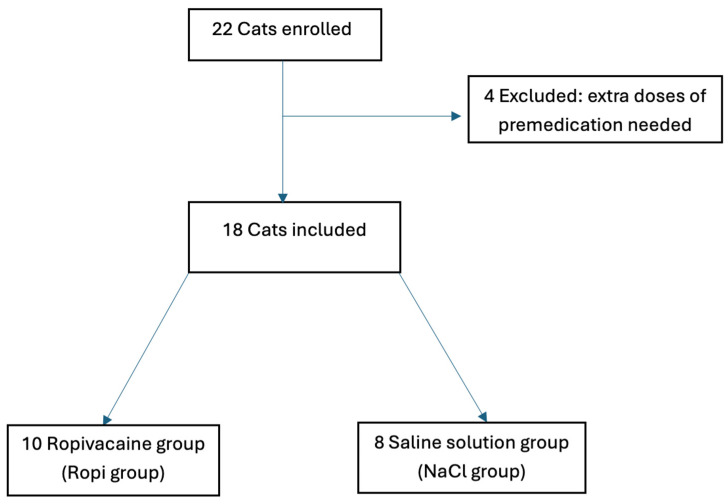
Study subject flow.

**Figure 2 vetsci-12-00524-f002:**
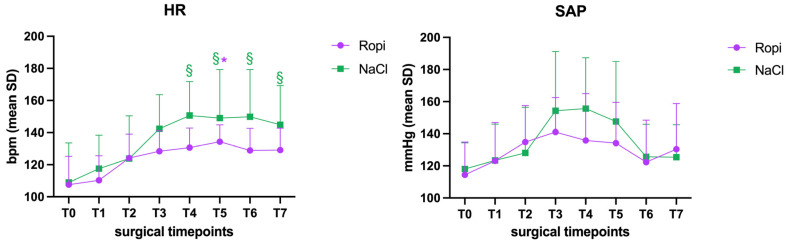
Mean values and standard deviation of heart rate (HR) and systolic arterial pressure (SAP) in the two groups: green NaCl and purple Ropi. * Significant difference in the Ropi group at T5 vs. T0; § significant difference in the NaCl group at T4, T5, T6 and T7 in comparison to T0.

**Figure 3 vetsci-12-00524-f003:**
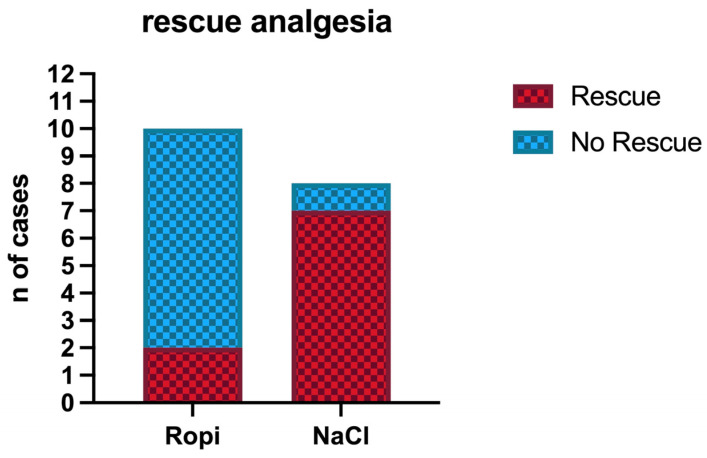
Needed of intraoperative rescue analgesia in the two groups.

**Table 1 vetsci-12-00524-t001:** Surgical timepoint.

Time	Procedure
T0	Five minutes prior to surgical draping
T1	Skin and subcutaneous tissue incision
T2	Looking for the first ovary
T3	ligation and removal of the first ovary
T4	Looking for the second ovary
T5	ligation and removal of the second ovary
T6	abdominal wall suture
T7	skin suture

**Table 2 vetsci-12-00524-t002:** Mean values and standard deviations of weight, age, duration of anesthesia, and duration of surgery in the two groups.

	Ropi Group	NaCl Group	*p*
Age (month)	11.75 ± 6.3	9.5 ± 2.4	0.39
Weight (kg)	2.9 ± 0.4	2.7 ± 0.4	0.28
Anesthesia duration (min)	118 ± 18	122 ± 26	0.63
Surgery duration (min)	55 ± 13	65 ± 14	0.12

**Table 3 vetsci-12-00524-t003:** Number of cats for each surgery time point that received bolus of fentanyl.

Time	Ropi Group	NaCl Group
T0	0/10	0/8
T1	1/10	2/8
T2	1/10	5/8
T3	0/10	0/8
T4	0/10	0/8
T5	0/10	0/8
T6	0/10	0/8
T7	0/10	0/8

## Data Availability

Data supporting the results stated above can be sent to anyone requesting them from the authors.

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
