# Peer review of "Low-Volume (0.3 mL/kg) Ropivacaine 0.5% for a Quadratus Lumborum Block in Cats Undergoing Ovariectomy: A Randomized Study"

_vetsci, 2025, doi:10.3390/vetsci12060524_

Round 1
Reviewer 1 Report (New Reviewer)
Comments and Suggestions for Authors
The paper submitted is an interesting study aiming to evaluate the effectiveness of the QLB (quadratus lumborum block) in cats undergoing laparotomic ovariectomy with a low volume of ropivacaine (0.3 mL/kg) in comparison to a control group.
The study was quite well performed and described, however there are some areas that could be improved. There are some minor inaccuracies in the manuscript text.
Specific points are listed below.
INTRODUCTION
-lines 60-61: “The transversus abdominis plane (TAP) block and the QLB are among…”
MATERIALS AND METHODS
-lines 106-110: Please rewrite the inclusion and exclusion criteria more clearly. The description of the criteria for choosing the population is a critical element of RCTs.
-line 107: What do the authors mean by "problematic behavior"?
-line 107-108: How do you exclude a patient a priori if he presents "irritation or infection of the injection site of the QLB"? These are manifestations that you highlight after the administration, therefore after he has been included in the study. In the case of infection, even a long time later...
-line 108: Please clarify what you mean by "unfavorable results"?
-lines 109-110: Please rephrase. The meaning of the sentence “Furthermore, all patients who required repeated the administration of premedication were excluded if were particularly agitated and refractory to the first dose” is unclear.
-line 115: arterial blood pressure? invasive or non-invasive? please specify.
-line 118: Please specify how the dosage of Ringer's lactate was adjusted based on the patient's needs. Otherwise delete the final part of the sentence.
-line 129: Please specify how the randomization was performed and the randomization ratio used.
-lines 170-172: Please describe the sample size calculation in more detail. Was the sample size calculated using software, formula, manual calculation, or other techniques? Please indicate the type of statistical test used to obtain the sample size. What was the primary outcome used in the reference study to identify the 84% effect? Was an effect size derived (please indicate type and data for the ES)?
-line 178: Tukey test.
-lines 177-178: To evaluate the trend over time, the ANOVA for repeated measures was rightly used. It is not clear why the one-way ANOVA for non-repeated data (+Tukey test) was used for the comparison between groups. In this case, there are 2 groups, so it is possible to simply use the t-test (as described later); the result should be similar, but I do not see the need to distinguish and use the ANOVA for a comparison between only 2 groups. An alternative general approach could be to use a two-way ANOVA considering both the group and time factors at the same time, but in this type of analysis even considering a trend analysis (within each group) separately from the comparison between groups is not to be considered wrong.
RESULTS
-figure 2: To have a more immediate graphic effect, I suggest referring to the group's identifying colours (e.g. green for the NaCl group and purple for the Ropi group), then distinguishing between the need for Rescue analgesia with a dotted effect or by lightening/darkening the same identifying colour of the group. I also suggest removing the title "fentanyl bolus" and perhaps inserting it in brackets in the caption "...rescue analgesya (fentanyl bolus)...". Another non-mandatory graphical suggestion could be to draw the graph in stacked bars, instead of interleaved bars.
DISCUSSION
-lines 289-291: It is recommended not to stress the limited number of cases as a limitation for this study, since a sample size calculation has been performed. Stressing the limited number of cases recruited when a sample size has been calculated a priori corresponds to criticizing the reliability of the method used by the authors themselves to calculate the sample size.
-lines 292-294: The fact that the procedures were conducted by students (moreover not one but "several students") increases the variability of the result and therefore must be rightly emphasized as a limitation of the study because it can be a great source of bias in the study. It is not clear why the study was carried out using surgical students... usually for this type of study, in order to reduce the source of bias deriving from the operator, it is preferable that the surgeon is, in addition to being experienced, always the same in all procedures.
CONCLUSIONS
lines 298-303: The conclusion of general effectiveness is a bit too optimistic. The main result of the study is the lower need for intraoperative rescue analgesia in the Ropi group compared to the NaCl group, and the conclusions should report this aspect and not a conclusion of effectiveness in an absolute sense.
Author Response
The paper submitted is an interesting study aiming to evaluate the effectiveness of the QLB (quadratus lumborum block) in cats undergoing laparotomic ovariectomy with a low volume of ropivacaine (0.3 mL/kg) in comparison to a control group.
The study was quite well performed and described, however there are some areas that could be improved. There are some minor inaccuracies in the manuscript text.
Thanks to the reviewer for the helpful comments. We have implemented the suggestions and revised the paper accordingly.
INTRODUCTION
-lines 60-61: “The transversus abdominis plane (TAP) block and the QLB are among…”
Lines 63: Corrected, thank you.
MATERIALS AND METHODS
-lines 106-110: Please rewrite the inclusion and exclusion criteria more clearly. The description of the criteria for choosing the population is a critical element of RCTs.
Lines 110-113: Clarified, as suggested.
-line 107: What do the authors mean by "problematic behavior"?
Lines 110-113: Thanks for point this out, we realized that the term was not correct.
-line 107-108: How do you exclude a patient a priori if he presents "irritation or infection of the injection site of the QLB"? These are manifestations that you highlight after the administration, therefore after he has been included in the study. In the case of infection, even a long time later...
Lines110-113: Patients were excluded if at the time of trichotomy and disinfection of the area dedicated to the execution of the block, an irritation/infection of the area was found. In that case the patients did not receive the nerve block but systemic analgesia and were consequently excluded from the study. Hopefully it is clearer now; thank you.
-line 108: Please clarify what you mean by "unfavorable results"?
Lines 110-113: we rephrase the sentences. Thank you.
-lines 109-110: Please rephrase. The meaning of the sentence “Furthermore, all patients who required repeated the administration of premedication were excluded if were particularly agitated and refractory to the first dose” is unclear.
Lines 110-113: Done, thank you.
-line 115: arterial blood pressure? invasive or non-invasive? please specify.
Line 118: Specified, thank you.
-line 118: Please specify how the dosage of Ringer's lactate was adjusted based on the patient's needs. Otherwise delete the final part of the sentence.
Line 123: Done, thank you.
-line 129: Please specify how the randomization was performed and the randomization ratio used.
Lines 132-133: Specified, thank you.
-lines 170-172: Please describe the sample size calculation in more detail. Was the sample size calculated using software, formula, manual calculation, or other techniques? Please indicate the type of statistical test used to obtain the sample size. What was the primary outcome used in the reference study to identify the 84% effect? Was an effect size derived (please indicate type and data for the ES)?
Lines 176-188: The sample size calculation was based on data from a previous study Lazzerini et al., in which the quadratus lumborum block (QLB) was effective in 84% of cats undergoing ovariectomy, as measured by the reduction in intraoperative rescue analgesia. The primary outcome used for the calculation was the proportion of cats not requiring intraoperative rescue analgesia, reflecting the block’s efficacy.
The calculation was performed using Clincalc.com. comparing a treatment efficacy of 84% in the QLB group to an estimated 20% in the control group based on clinical experience and literature benchmarks; we estimated the sample size.
With an alpha error of 0.05 and 80% power, the software indicated that a minimum of 8 cats per group would be sufficient to detect a statistically significant difference. To account for potential dropouts or protocol deviations, we increased the group size to 11 cats.
-line 178: Tukey test.
Line 185, corrected, thank you.
-lines 177-178: To evaluate the trend over time, the ANOVA for repeated measures was rightly used. It is not clear why the one-way ANOVA for non-repeated data (+Tukey test) was used for the comparison between groups. In this case, there are 2 groups, so it is possible to simply use the t-test (as described later); the result should be similar, but I do not see the need to distinguish and use the ANOVA for a comparison between only 2 groups. An alternative general approach could be to use a two-way ANOVA considering both the group and time factors at the same time, but in this type of analysis even considering a trend analysis (within each group) separately from the comparison between groups is not to be considered wrong.
Lines 183-188: thank you to point out this aspect, we corrected the sentence as suggested.
RESULTS
-figure 2: To have a more immediate graphic effect, I suggest referring to the group's identifying colours (e.g. green for the NaCl group and purple for the Ropi group), then distinguishing between the need for Rescue analgesia with a dotted effect or by lightening/darkening the same identifying colour of the group. I also suggest removing the title "fentanyl bolus" and perhaps inserting it in brackets in the caption "...rescue analgesya (fentanyl bolus)...". Another non-mandatory graphical suggestion could be to draw the graph in stacked bars, instead of interleaved bars.
Figure 2: Modified as suggested, thank you.
DISCUSSION
-lines 289-291: It is recommended not to stress the limited number of cases as a limitation for this study, since a sample size calculation has been performed. Stressing the limited number of cases recruited when a sample size has been calculated a priori corresponds to criticizing the reliability of the method used by the authors themselves to calculate the sample size.
Lines 302-304: Thank you for the suggestion, we modify the text accordingly.
-lines 292-294: The fact that the procedures were conducted by students (moreover not one but "several students") increases the variability of the result and therefore must be rightly emphasized as a limitation of the study because it can be a great source of bias in the study. It is not clear why the study was carried out using surgical students... usually for this type of study, in order to reduce the source of bias deriving from the operator, it is preferable that the surgeon is, in addition to being experienced, always the same in all procedures.
We already included in the limitation of the study even if the use of students in this study was intentional in order to simulate real-world conditions where the procedure is not always performed by a single expert. In clinical settings, basic surgeries are often conducted by different personnel at varying levels of experience. By involving students, we aimed to assess the effectiveness of the block under more typical circumstances, where variability in operator skill is present.
CONCLUSIONS
lines 298-303: The conclusion of general effectiveness is a bit too optimistic. The main result of the study is the lower need for intraoperative rescue analgesia in the Ropi group compared to the NaCl group, and the conclusions should report this aspect and not a conclusion of effectiveness in an absolute sense.
Lines 313- 318: Thank you for the suggestion, we modified the conclusions as suggested by all the reviewers.
Reviewer 2 Report (New Reviewer)
Comments and Suggestions for Authors
Reviewer Comments to the Author – Minor Revision
General comments:
This manuscript presents a well-executed, randomized, blinded controlled study assessing the analgesic efficacy of a low-volume (0.3 mL/kg) 0.5% ropivacaine QLB in cats undergoing elective ovariectomy. The study is timely and adds valuable information to a field where clinical data are still limited. While the experimental design and statistical analysis are appropriate, the novel aspects and differences from existing literature could be emphasized more clearly, especially in the Introduction and Discussion. With minor revisions aimed at clarifying the manuscript’s contribution to the field, the article will be suitable for publication.
Specific Comments:
1. Clarification of Novelty and Comparison to Existing Studies
-
Introduction (Lines 85–97):
The authors mention that previous studies in cats used higher volumes of local anesthetic (e.g., 0.4–0.5 mL/kg of bupivacaine or ropivacaine). However, the novelty of using low-volume 0.3 mL/kg of 0.5% ropivacaine should be more explicitly framed—particularly given that a previous study using 0.3 mL/kg bupivacaine showed poorer analgesic outcomes. Please emphasize that your study contradicts the notion that low volume is ineffective, potentially due to the higher concentration used. A sentence such as the following would help:“Contrary to prior findings where a low volume of 0.3 mL/kg bupivacaine led to increased analgesic needs [20], this study demonstrates that using the same volume of 0.5% ropivacaine achieves effective intraoperative analgesia, suggesting that concentration may compensate for lower volume.”
-
Discussion (Lines 266–284):
While the study is compared to previous work (e.g., Lazzarini et al., Paolini et al., dos Santos et al.), the distinct methodological differences (volume, concentration, use of ketamine, surgeon experience) should be more directly highlighted. I suggest adding a paragraph that compares these variables explicitly and states how the present study fills a unique gap—for example:“To our knowledge, this is the first study to demonstrate that a 0.3 mL/kg dose of high-concentration (0.5%) ropivacaine is sufficient to provide intraoperative analgesia comparable to higher volume protocols, thus supporting a more volume-conservative approach in clinical feline anesthesia.”
2. Additional Specific Points
-
Abstract (Lines 25–42): Please consider briefly mentioning the contrast between your findings and previous low-volume studies in the abstract to highlight the novelty.
-
Methods (Lines 102–104): It may be useful to explicitly state that no other study to date has used this specific combination of volume, concentration, and drug (ropivacaine 0.5% at 0.3 mL/kg) in a clinical feline model with blinded observers.
-
Conclusion (Lines 298–303): Consider concluding with a more assertive statement about the potential clinical value of your low-volume protocol (e.g., reducing local anesthetic toxicity risk while maintaining efficacy).
Conclusion and Recommendation
This study is a clinically relevant advancement in feline regional anesthesia, particularly in minimizing drug volume without sacrificing analgesic efficacy. The data are strong and the study is well controlled. To maximize its impact, I recommend minor revisions with a focus on more clearly articulating how this study diverges from and builds upon previous research.
Author Response
Reviewer 2
Reviewer Comments to the Author – Minor Revision
General comments:
This manuscript presents a well-executed, randomized, blinded controlled study assessing the analgesic efficacy of a low-volume (0.3 mL/kg) 0.5% ropivacaine QLB in cats undergoing elective ovariectomy. The study is timely and adds valuable information to a field where clinical data are still limited. While the experimental design and statistical analysis are appropriate, the novel aspects and differences from existing literature could be emphasized more clearly, especially in the Introduction and Discussion. With minor revisions aimed at clarifying the manuscript’s contribution to the field, the article will be suitable for publication.
We thank the reviewer for their valuable suggestions, which helped us refine and improve our work.
Specific Comments:
- Clarification of Novelty and Comparison to Existing Studies
- Introduction (Lines 85–97):
The authors mention that previous studies in cats used higher volumes of local anesthetic (e.g., 0.4–0.5 mL/kg of bupivacaine or ropivacaine). However, the novelty of using low-volume 0.3 mL/kg of 0.5% ropivacaine should be more explicitly framed—particularly given that a previous study using 0.3 mL/kg bupivacaine showed poorer analgesic outcomes. Please emphasize that your study contradicts the notion that low volume is ineffective, potentially due to the higher concentration used. A sentence such as the following would help:
“Contrary to prior findings where a low volume of 0.3 mL/kg bupivacaine led to increased analgesic needs [20], this study demonstrates that using the same volume of 0.5% ropivacaine achieves effective intraoperative analgesia, suggesting that concentration may compensate for lower volume.”
Lines 92-97: Corrected as suggested, thank you.
- Discussion (Lines 266–284):
While the study is compared to previous work (e.g., Lazzarini et al., Paolini et al., dos Santos et al.), the distinct methodological differences (volume, concentration, use of ketamine, surgeon experience) should be more directly highlighted. I suggest adding a paragraph that compares these variables explicitly and states how the present study fills a unique gap—for example:
“To our knowledge, this is the first study to demonstrate that a 0.3 mL/kg dose of high-concentration (0.5%) ropivacaine is sufficient to provide intraoperative analgesia comparable to higher volume protocols, thus supporting a more volume-conservative approach in clinical feline anesthesia.”
Lines 297-301: Implemented as suggested, thank you.
- Additional Specific Points
- Abstract (Lines 25–42): Please consider briefly mentioning the contrast between your findings and previous low-volume studies in the abstract to highlight the novelty.
Lines 39-44: added, thank you.
- Methods (Lines 102–104): It may be useful to explicitly state that no other study to date has used this specific combination of volume, concentration, and drug (ropivacaine 0.5% at 0.3 mL/kg) in a clinical feline model with blinded observers.
Lines 142-144: Implemented as suggested, thank you.
- Conclusion (Lines 298–303): Consider concluding with a more assertive statement about the potential clinical value of your low-volume protocol (e.g., reducing local anesthetic toxicity risk while maintaining efficacy).
Lines 313-318 :Added as suggested, thank you.
Conclusion and Recommendation
This study is a clinically relevant advancement in feline regional anesthesia, particularly in minimizing drug volume without sacrificing analgesic efficacy. The data are strong and the study is well controlled. To maximize its impact, I recommend minor revisions with a focus on more clearly articulating how this study diverges from and builds upon previous research.
Round 2
Reviewer 1 Report (New Reviewer)
Comments and Suggestions for Authors
I thank the Editor for giving me the opportunity to review this interesting paper and I thank the Authors for making the requested changes. The current version of the manuscript can be considered suitable for publication.
This manuscript is a resubmission of an earlier submission. The following is a list of the peer review reports and author responses from that submission.
Round 1
Reviewer 1 Report
Comments and Suggestions for Authors
General: The manuscript investigates the use of a QLB in cats undergoing ovariectomy. While the topic contributes to the current literature, several flaws need to be addressed before it can be considered for publication.
The dose and model used are similar to a previous study published in 2024, which utilized 3.2mg/kg in a larger sample size of cats undergoing ovariectomy. Why do the authors aim to reduce the volume of the analgesic in question? The article lacks a hypothesis or explanation regarding this choice. Why do the authors intend to repeat the same model that was previously published? A better justification is required. Additionally, the conclusion is not supported by the results.
Other suggestions:
Title: low volume or low dose? what is the goal of the paper volume or dose? Remove the amount for the title
The simple summary is confusing and should be rewritten; it is hard to follow and is not concise.
Lines 16-17: the rescue protocol description is confusing
Line 17: add coma after surgery. Clinical parameters?
Abstract: The core elements of the research are not concisely presented and need to be rewritten. Do not include the exclusion criteria; exclude the word “we.” if reading the abstract, the reader does not know what the T5 or T0 means.
Line 38: conclusion: is it effective in pain management? Is it effective in cats undergoing ovariectomy?
Key words: remove words that are included in the title – quadratus lumborum block, remove the repeated word anesthesia
Introduction: start with the importance of pain control and not with the surgical procedure. The suggestion is to start with the evolution of ERA, which started in 1997 in human medicine and in 2022 in veterinary medicine.
line 46: what do other systems conditions mean?
Line 54: Substitute are for is
Lines 53-56: Describe and make it more clear the difference between TAP and QLB blocks. The way it is described in the introduction does not explain the difference well
Lines 58-69: the authors should rewrite the mechanism of action and the hypothesis
Lines 70-73 and 74 - 78: merge the 2 paragraphs.
Include why the ovariectomy procedure is a good model to test the efficacy of the block
Material and methods:
Lines 85-89: sample size estimate should be included in the statistical analysis section
Lines 101: is the dexmed dose correct? 15 micrograms/kg?
Line 116: Why was prophylactic antibiotic given to the patients if it was an elective surgery?
Line 119: what random.org mean?
Suggestions: Include if surgery was performed by the same doctor or different doctors.
Was post-op pain accessed? If yes which pain scale was used?
Was the extubation time recorded?
Results:
Lines 176 – 180: My understanding is the HR and other parameters did not differ between the groups. If that is correct, why was there a need for fentanyl doses intraoperatively? And atropine.
Suggestions: include a table per animal with the anesthesia parameter and the time fentanyl was given.
Include the times when a fentanyl bolus was administered.
Include the atropine doses and extubation time per animal.
Include the total anesthetic time from induction to extubation and how these cats recovered (dysphoric or not).
Discussion:
The results do not support the conclusion since the results did not demonstrate when fentanyl bolus was given, there was no difference between groups, and atropine was given to many patients.
Comments on the Quality of English Language
The text should be proofread by a native English speaker for clarity and correctness.
Author Response
General: The manuscript investigates the use of a QLB in cats undergoing ovariectomy. While the topic contributes to the current literature, several flaws need to be addressed before it can be considered for publication.
The dose and model used are similar to a previous study published in 2024, which utilized 3.2mg/kg in a larger sample size of cats undergoing ovariectomy. Why do the authors aim to reduce the volume of the analgesic in question? The article lacks a hypothesis or explanation regarding this choice. Why do the authors intend to repeat the same model that was previously published? A better justification is required. Additionally, the conclusion is not supported by the results.
Other suggestions:
Title: low volume or low dose? what is the goal of the paper volume or dose? Remove the amount for the title
The simple summary is confusing and should be rewritten; it is hard to follow and is not concise.
We sincerely appreciate the time and effort the reviewer has dedicated to evaluating our work, as their insights and feedback are invaluable in enhancing the overall quality of our research. We have also revised the English as suggested.
The goal of our work was to verify whether the use of a low volume of ropivacaine for quadratus lumborum block could be effective in intraoperative analgesic management in cats.
We rewritten the simple summary as suggested (lines 11-24).
Lines 16-17: the rescue protocol description is confusing
A more comprehensive explanation is included in the text (lines 340-345), and since the brief summary is limited to 250 words, we opted not to elaborate further.
Line 17: add coma after surgery. Clinical parameters? Corrected as suggested.
Abstract: The core elements of the research are not concisely presented and need to be rewritten. Do not include the exclusion criteria; exclude the word “we.” if reading the abstract, the reader does not know what the T5 or T0 means. We have rewritten the abstract as suggested (line 25-42).
Line 38: conclusion: is it effective in pain management? Is it effective in cats undergoing ovariectomy? Corrected, thank you for the suggestions.
Key words: remove words that are included in the title – quadratus lumborum block, remove the repeated word anesthesia Done, thank you, line 130.
Introduction: start with the importance of pain control and not with the surgical procedure. The suggestion is to start with the evolution of ERA, which started in 1997 in human medicine and in 2022 in veterinary medicine. Edited as suggested, lines 137-145.
line 46: what do other systems conditions mean? Specify as suggested, lines 135-137.
Line 54: Substitute are for is Corrected, thank you.
Lines 53-56: Describe and make it more clear the difference between TAP and QLB blocks. The way it is described in the introduction does not explain the difference well Edited as suggested, lines 147-157.
Lines 58-69: the authors should rewrite the mechanism of action and the hypothesis Done, thank you.
Lines 70-73 and 74 - 78: merge the 2 paragraphs. Edited as suggested.
Include why the ovariectomy procedure is a good model to test the efficacy of the block
Done, thank you for the suggestion. Lines 174-178.
Material and methods:
Lines 85-89: sample size estimate should be included in the statistical analysis section
Corrected, thank you lines 361-364.
Lines 101: is the dexmed dose correct? 15 micrograms/kg? Yes, the dosage of dexmedetomidine is correct.
Line 116: Why was prophylactic antibiotic given to the patients if it was an elective surgery? In the discussions section we reported that the surgeries were conducted by non-expert surgeons, which may have extended the surgical duration of the procedure and heightened the risk of pathogen exposure. Consequently we decided to administer a single dose of antibiotics for prophylactic purposes.
Line 119: what random.org mean? random.org is the website we used for randomization (We added in the text, line 299).
Suggestions: Include if surgery was performed by the same doctor or different doctors. Added, thank you for the suggestions, lines 339-340.
Was post-op pain accessed? If yes which pain scale was used? Post-operative pain assessment could not be conducted since the cats were discharged within hours of the surgery after their health status was confirmed to be satisfactory.We added in the text (lines 357-359).
Was the extubation time recorded? Thank you for the suggestion. The duration of anesthesia was determined by measuring the time from the induction to the extubation process, we specify in the text lines 355-356.
Results:
Lines 176 – 180: My understanding is the HR and other parameters did not differ between the groups. If that is correct, why was there a need for fentanyl doses intraoperatively? And atropine.
The heart rate in the Ropi group was significantly higher at time T5 compared to T0 (p= 0.03). In contrast, the NaCl group exhibited a significant increase from T0 at times T4 to T7 (p<0.01) (Figure 1). However, other parameters, including arterial blood pressure, temperature, capillary refill time (CRT), ECG rhythm, end-tidal carbon dioxide (EtCO2), oxygen saturation (SpO2), fractional isoflurane concentration (Fe’Iso), and spirometry, did not reveal any significant differences either within the groups or between them.
Suggestions: include a table per animal with the anesthesia parameter and the time fentanyl was given. Added the table in text as suggested, thank you Lines 406-409
Include the times when a fentanyl bolus was administered. Done, thank you.
Include the atropine doses and extubation time per animal. We included the dosage thank you.
Include the total anesthetic time from induction to extubation and how these cats recovered (dysphoric or not). The total anesthetic time was already in the text (table 2, line 384), we added how these cats recovered, thank you for the suggestions ( lines 380-381).
Discussion:
The results do not support the conclusion since the results did not demonstrate when fentanyl bolus was given, there was no difference between groups, and atropine was given to many patients.
We have implemented the discussions section, we hope that the work is now ready for publication.
Reviewer 2 Report
Comments and Suggestions for Authors
The manuscript “Low Volume (0.3 mL/kg) Ropivacaine 0.5% for a Quadratus Lumborum Block, in Cats Undergoing Ovariectomy:
A Randomized Study” it is an interesting and well-structured work considering the objectives. Some minor recommendations:
Abstract: Define acronymous: HR, T0, T5....Attach main conclusions
Introduction: Add the null hypothesis information
Line 178: Define “other parameters” and present normal results
Statistical analysis – Are you performed normal distribution tests? Why you choose parametric tests for statistical analysis?
Why didn't you do a comparison between groups (Hopi versus NaCl)
What was your null hypothesis? Was it accepted or rejected?
References list you can improve, some recent were missed.
Author Response
The manuscript “Low Volume (0.3 mL/kg) Ropivacaine 0.5% for a Quadratus Lumborum Block, in Cats Undergoing Ovariectomy:
A Randomized Study” it is an interesting and well-structured work considering the objectives. Some minor recommendations:
We appreciate the reviewer's efforts in assisting us to enhance the quality of our work.
Abstract: Define acronymous: HR, T0, T5....Attach main conclusions.
Done, thank you for the suggestions.
Introduction: Add the null hypothesis information. Added in the texts, thank you.
Line 178: Define “other parameters” and present normal results Done, thank you.
Statistical analysis – Are you performed normal distribution tests? Why you choose parametric tests for statistical analysis? The data were analyzed for normal distribution using a D’Agostino and Pearson tests and are expressed by mean and standard deviation Lines 365-373.
Why didn't you do a comparison between groups (Hopi versus NaCl) We compared the 2 groups, and we did not highlighted any differences, we added a sentence regarding HR in the text.
What was your null hypothesis? Was it accepted or rejected? Our initial hypothesis posited that administering a low volume of 0.3 ml/kg in conjunction with a high concentration of ropivacaine at 0.5% could serve as an effective method for managing nociception during the intraoperative period. This hypothesis was subsequently validated through our findings, which indicated that the cats receiving ropivacaine demonstrated a reduced requirement for fentanyl compared to the patients that was administered a saline blockade.
References list you can improve, some recent were missed. Added, thank you for the suggestions.
Reviewer 3 Report
Comments and Suggestions for Authors
This is an interesting paper which clearly indicates the advantage of QLB technique vs TAP technique. However the novelity of the project is rather moderate. Major criticism: in the title authors write "low volume (0.3 mL/kg) ropivacaine" and in the abstract "3 mg/kg ropivacaine". It is a little bit confusing. Also conclusion is absolutely too short. It has to be changed. In general before publishing a major revisions have to be performed.
Author Response
This is an interesting paper which clearly indicates the advantage of QLB technique vs TAP technique. However the novelity of the project is rather moderate. Major criticism: in the title authors write "low volume (0.3 mL/kg) ropivacaine" and in the abstract "3 mg/kg ropivacaine". It is a little bit confusing. Also conclusion is absolutely too short. It has to be changed. In general before publishing a major revisions have to be performed.
We would like to thank the reviewer's time to help us implement the quality of our work.We concur with the reviewer's observations and have revised the text to eliminate any ambiguity between 3 mg/kg and 0.3 ml/kg. The aim of our study was to assess the efficacy of combining a low volume of 0.3 ml/kg with a high concentration of ropivacaine at 0.5% in the quadratus lumborum block for pain management in cats undergoing oophorectomy. We have expanded the conclusions as suggested, we hope that the paper is now ready for publication
Round 2
Reviewer 1 Report
Comments and Suggestions for Authors
Comments are attached

The manuscript needs to be proofread by a native English speaker. There are several grammatical and punctuation errors that must be corrected before it is suitable for publication.
Author Response
Reviewer 1
Major considerations
The manuscript needs to be proofread by a native English speaker. There are several grammatical and punctuation errors that must be corrected before it is suitable for publication.
The study has been carefully revised for grammatical point of view and we hope it is now ready for publication.
The authors did not explain why they aimed to reduce the volume of ropivacaine in QLB. What is the advantage of doing this?
There are multiple articles published providing enough information on the efficacy of QLB using ropivacaine in visceral pain. If their goal is to reduce the volume, they need to state the reason/advantages behind that.
The possibility of using a smaller volume of local anesthetic allows to obtain several advantages such as the possibility of increasing the concentrations (reducing the dilution), reducing the possibility of toxicity, and allowing an equipotent analgesic level. These concepts are widely reported in the litterature, and we believe that the hypothesis and the aim of our study have been widely discussed.
A statistician should be consulted for sample size and better analysis to be sure there is a difference or not.
The statistical analysis and the calculation of the sample size were carefully performed by a colleague with decades of experience in the field of statistics. We therefore believe that consulting with an additional specialist is not necessary; thank you for the suggestion.
Other considerations
Simple summary
Line 16, 17: "In instances of pain, a fentanyl bolus was provided, or an infusion was initiated if the bolus was insufficient." The authors need to clarify if the infusion (IV, IM. SC) was done with fentanyl or another drug after the fentanyl bolus.
Corrected as requested, thank you [lines 16-17].
Lines 25 -26 Change the order: "The purpose of this randomized prospective study was to assess the effectiveness of a low volume of ropivacaine 0.5% (0.3 mL/kg) for performing a transverse quadratus lumborum block (QLB) in cats undergoing ovariectomy. "
Corrected, thank you [lines25-27].
Line 32: anesthesia was maintained with isoflurane - replace maintenance. What was the dose for induction and iso maintenance? Corrected as suggested [line 31-32].
Introduction:
Line 48: replace breast with mammary. Done, [line 48].
Line 61, 88: Do not repeat the word, only the acronym QLB (first used on line 25). Review the entire text. Reviewed the entire text, as suggested, thank you.
Lines 95-99: the authors do not need to keep the hypothesis and the study goal. Suggestion: "This study aimed to evaluate the efficacy of a low volume (0.3 mL/kg) of ropivacaine 0.5% in reducing intraoperative analgesia requests in cats undergoing ovariectomy." We choose to retain the hypothesis and the study objective, as recommended by another reviewer.
Material and methods:
The review methods description is confusing and does not follow an order
Line 26: What did the authors mean by "When possible"? Or was it included or not in the anesthesia protocol? The arterial catheter was placed during the preparation of the animal, before entering the operating room. We have corrected and clarified as suggested, thank you.
"A second venous catheter was then positioned for the possible administration of drugs during surgery" - rewrite - substitute the highlighted text. We typically positioned two vascular access: one for continuous infusions while reserving the other for potential emergency drug administration, such as atropine. This has been clearly outlined in the text [lines 126-128].
Line 133: is the concentration and drug description correct? What is the drug concentration used in the study? Ropivacaine (Ropivacaine Kabi 7.5 mg/mL, Fresenius Kabi, Italy) at 0.5%. Do the authors mean to write ropivacaine 0.5% 5mg/ml or 7.5% 0.75mg/ml. The drug used in our study was the Ropivacaine Kabi 7.5 mg/mL, Fresenius Kabi, Italy and we diluted it in order to reach a concentration of 0.5%
Line 164: complete the sentence "In case of hypotension (MAP < 60 mmHg) with concurrent bradycardia (HR ≤ 100bpm), atropine at the dosage of 0.02 mg kg-1 IV (atropine sulfate, A.T.I., Bologna, Italy). Corrected [lines 162-164].
Line 177: one group has 10 and the control has 8, not 11 each.
This is reported in the results. In the statistics section, it was explained how the sample size was determined. Specifically, the minimum number was 8 patients per group, and we decided to increase it to 11 per group to account for potential dropouts. In the end, after excluding 4 patients due to the need to re-administer premedication, the final number of patients was 10 in the ropi group and 8 in the saline group, both of which are valid numbers for a proper statistical analysis.
Line 197: the figure legend, states a significant increase compared to TO from time T4 to time 17, there is only one * in the graphic. Furthermore, the data variation is big and the sample size small to be able to conclude there is a difference. Recommend a consultation with a statistician.
Dear reviewer, we understand your point but, the sample size was calculated to evaluate the efficacy of the block and we are aware that other differences between the 2 groups could not have been discovered, but the differences you are speaking about are inside each group and if the statistical test highlighted a significance this is true regardless the low number of animals. We modified the legend because maybe it was not clear line 202-203)
Lines 219-221: when the atropine was given? The time points should be presented.
Thanks for the suggestion, we have reported the times when atropine was administered. [lines 218-220].
Discussion: The discussion is confusing and should be rewritten; it is hard to follow and is not concise.
Dear reviewer we corrected the discussion following also the indication of the other reviewers and it is not possible to reduce it.
Line 224: use QLB acronym Done, thank you [line 224].
Lines 228-231: this belongs to the results. "From a hemodynamic point of view, a significant increase in heart rate was detected at time TS (ligation and removal of the second ovary) compared to time TO (pre-surgery) in the ropivacaine group. In the NaCl group, a significant increase in heart rate was observed from time T4 (search for the second ovary) to time T7 (skin suturing) compared to time TO (pre-surgery)*.
We agree and have indeed included this in the results section; however, to facilitate the reader's understanding of the subsequent discussion, we have reiterated the point.
Line 232: The use of a high dose of dexmedetomidine could be causing severe bradycardia, and this might be why atropine was needed in almost all cats, regardless of the group. It would be interesting to include in the discussion the drugs used in the multimodal protocol because that will interfere with the hemodynamic changes for the patients.
The use of dexmedetomidine in our study was associated with the use of alfaxalone both in premedication and in induction. Since the tachycardic effect of alfaxalone is known, the use of 15mcg/kg of dexmedetomidine should not significantly impact the patient from a cardiovascular point of view.
Line 236-239: Why do the authors say that the use of atropine can be the cause of the increased heart rates in the QLB group and not in the control group? Since it was not in the results when the patients received atropine, it is hard to discuss or conclude that.
Added, thank you [lines 236-239].
Line 286: the authors did not include any evaluation or describe the surgeons in the methods sections. It was not part of the purpose of the study, so no discussion was included.
What do novice surgeons entitle? Students, new grads? Does it mean that tissue handling was not adequate, and surgical time was extended? Needs more clarification on the methods if it wants to be included in the discussion section.
The surgeons were students, and this has been specified, thanks for the suggestion. The timing of surgery was in line with what is usually see in clinical practice (the students had undergone specific training before performing the surgery).
Include a table per animal with all the anesthesia parameters and the time fentanyl and other meds were given. We have already responded and amended this in the previous review regarding time fentanyl. However, we do not think that adding a further table with the administration of atropine is necessary.
Including the times when a fentanyl bolus was administered. We have already responded and amended this in the previous review.
Including the atropine doses and extubation time per animal. Done, thank you.
Include the total anesthetic time from induction to extubation and how these cats recovered (dysphoric or not). We have already responded and amended this in the previous review.
Reviewer 3 Report
Comments and Suggestions for Authors
Thanks for all your work to improve the final form of the manuscript. Now it looks much better and I agree for its publication in the present form.
Author Response
Thank you for taking the time to review our work.